# Role of air-sea heat flux on the transformation of Atlantic Water encircling the Nordic Seas

Jie Huang [1] ✉, Robert S. Pickart [1], Zhuomin Chen[2] & Rui Xin Huang[1]

The warm-to-cold densification of Atlantic Water (AW) around the perimeter of the Nordic Seas is a critical component of the Atlantic Meridional Overturning Circulation (AMOC). However, it remains unclear how ongoing changes in air-sea heat flux impact this transformation. Here we use observational data, and a one-dimensional mixing model following the flow, to investigate the role of air-sea heat flux on the cooling of AW. We focus on the Norwegian Atlantic Slope Current (NwASC) and Front Current (NwAFC), where the primary transformation of AW occurs. We find that air-sea heat flux accounts almost entirely for the net cooling of AW along the NwAFC, while oceanic lateral heat transfer appears to dominate the temperature change along the NwASC. Such differing impacts of air-sea interaction, which explain the contrasting long-term changes in the net cooling along two AW branches since the 1990s, need to be considered when understanding the AMOC variability.

The water mass transformation that occurs in the Nordic Seas plays a critical role in regulating the Atlantic Meridional Overturning Circulation (AMOC), which helps maintain Earth's Climate[1–5]. In addition to the open-ocean convection in the interior of the Greenland and Iceland Seas[6,7], the warm-to-cold transformation of Atlantic Water (AW) as it circulates around the perimeter of the Nordic Seas (so called the rim current transformation) is a key process producing dense water in the North Atlantic[8,9]. Air–sea heat flux (the ocean heat loss to the atmosphere) was initially considered the major driver for the transformation of AW in the Nordic Seas[8]. Later, regional studies demonstrated the important role of lateral oceanic processes (e.g., advection and mixing of heat) on this rim current transformation[10–14]. For instance, by analyzing simultaneous observations of the atmosphere and ocean during a cold-air outbreak in the western Iceland Sea, the ocean's cooling response to intensified air–sea heat loss in the boundary current region was found to be offset by lateral processes[14]. In addition, high-resolution numerical model studies have suggested a comparable role of oceanic lateral processes compared to air–sea heat flux[15,16]. However, these models are unable to accurately represent the Norwegian Atlantic Front Current, one of the major conduits of AW in the Nordic Seas. To date, it remains unclear how much of the

transformation along the rim current system in the Nordic Seas is directly attributable to air–sea heat flux, especially for the Norwegian Atlantic Front Current.

Since the first work emphasizing the importance of the rim current transformation of AW in the 1990s[8], rapid changes have been taking place in the Nordic Seas: the wintertime sea ice has retreated;[17] the poleward heat transport across the Greenland-Scotland Ridge[4] and equatorward freshwater flux through Fram Strait[18] have increased; and the ocean heat loss to the atmosphere has declined[19]. These changes in recent decades have had pronounced impacts on the water mass transformation in this region[20]. For example, studies have argued that the reduction of ocean heat loss has weakened deep convection in the Greenland and Iceland Seas[19,21,22]. In addition, the retreat of wintertime sea ice has led to enhanced ventilation of AW as it flows southward in the East Greenland Current[17]. Considering the large-scale ramifications of these changes—e.g., the cooling of the North Atlantic warming hole driven by the changes in the high-latitude heat transport[3] and the reorganization of air–sea interaction due to the sea-ice retreat[23]—it is of high interest to further examine the rim current transformation of AW.

The rim current system considered in this study consists of three boundary currents: the two branches of the northward-flowing

[1]Woods Hole Oceanographic Institution, Woods Hole, MA, USA. [2]Department of Marine Sciences, University of Connecticut, Groton, CT, USA. ✉e-mail: jhuang@whoi.edu

Norwegian Atlantic Current (NwAC) in the eastern Nordic Seas, referred to as the Norwegian Atlantic Slope current (NwASC) and the Norwegian Atlantic Front Current (NwAFC); and the southward flowing East Greenland Current (EGC) along the continental slope of Greenland (see Fig. 1a). Since most of the warm-to-cold transformation of AW occurs in the Norwegian Sea, we focus mainly on the two NwAC branches. After identifying the branches, we use historical hydrographic data to quantify the along-pathway evolution of the AW. By applying a one-dimensional mixing model following the NwASC and NwAFC, we assess the role of air–sea heat flux on the transformation within each branch from a Lagrangian perspective. Finally, we investigate the long-term variability in the rim current transformation of AW and how this is related to changes in air–sea heat flux in the Nordic Seas.

## Results

### Pathways of Atlantic Water along the rim current system

We identified the AW pathways using the long-term mean sea surface velocity fields from satellite observations and a reanalysis product (described in the "Methods" section). In particular, the AW pathways correspond to the maxima in the composite long-term mean velocity field from the two products (see Fig. 1b; the maxima are indicated by the solid lines). The pathways so identified are consistent with previously schematized depictions of the rim current system in the Nordic Seas (see also Fig. 1a). We define the boundary current region as a 50 km swath centered on the velocity maxima (indicated by the black dashed lines in Fig. 1b), in line with past studies[24–26]. Note that we do not consider the transition pathway (indicated by the purple dashed line in Fig. 1b) that connects the eastern and western boundary currents, due to the large variability of velocity field in this region. We first focus on the mean state from 2005 to 2018.

In addition to the surface velocity field, we also used a metric known as the $\sigma_0$-$\pi_0$ distance to investigate the pathways of the AW, where $\sigma_0$ and $\pi_0$ are the potential density and potential spicity, respectively, referenced to the sea surface. We applied the metric to a recently compiled historical hydrographic dataset for the Nordic Seas from 1993 to 2018 (see "Methods" section for details of the dataset and

metric). By calculating the $\sigma_0$-$\pi_0$ distance between any water parcel in the Nordic Seas and a characteristic dense mode of AW ($\sigma_0$ = 28.0 kg m$^{-3}$, $\pi_0$ = −2.98 kg m$^{-3}$, see Supplementary Fig. 1 for the identification of this mode), potential pathways of AW emerge from the distribution map of small distance (<0.05 kg m$^{-3}$) between 550 and 600 m (Fig. 1c). These pathways are consistent with those in the long-term mean sea surface velocity field. We note that 550–600 m is the only depth range that the distribution of small $\sigma_0$-$\pi_0$ distance resembles the circulation map of Fig. 1a. The AW located deeper than this is isolated within the interior basins (e.g., the Lofoten Basin), while the AW located above is significantly modified as it flows along the rim current system. As such, the depth of 550 m, whose properties are relatively conserved, represents the lower boundary of the portion of the AW layer as it circulates around the Nordic Seas.

### Transformation of Atlantic Water along the rim current pathways

We assembled all hydrographic profiles within the 50 km swath corresponding to the surface velocity pathways and constructed composite vertical sections of properties along each pathway. The vertical sections of potential temperature for the three boundary currents are shown in Fig. 2a–c, with the along-stream direction indicated. Our focus is on the AW layer within the upper 550 m of water column, hence the Polar Surface Water (PSW) in the upper layer of the NwASC and EGC ($S$ < 34.5 and $\sigma_0$ < 27.7 kg m$^{-3}$) is masked by gray shading in Fig. 2a, c. The cooling of AW along the direction of flow is evident, especially for the two branches of the NwAC. To quantify this, we computed the depth-mean potential temperature and density over the AW layer following each branch (Fig. 2d, e). This reveals that the densification of AW layer primarily occurs along the NwASC and NwAFC in the eastern Nordic Seas, dominated by the cooling in temperature (rather than the change in salinity, see Supplementary Fig. 2). This is consistent with previous studies, who hypothesized that the cooling of AW is primarily driven by ocean heat loss to the atmosphere[8].

By contrast, there is comparatively less cooling of the AW along the EGC, and no net change in density due to the fact that changes in temperature and salinity compensate each other along this branch (see

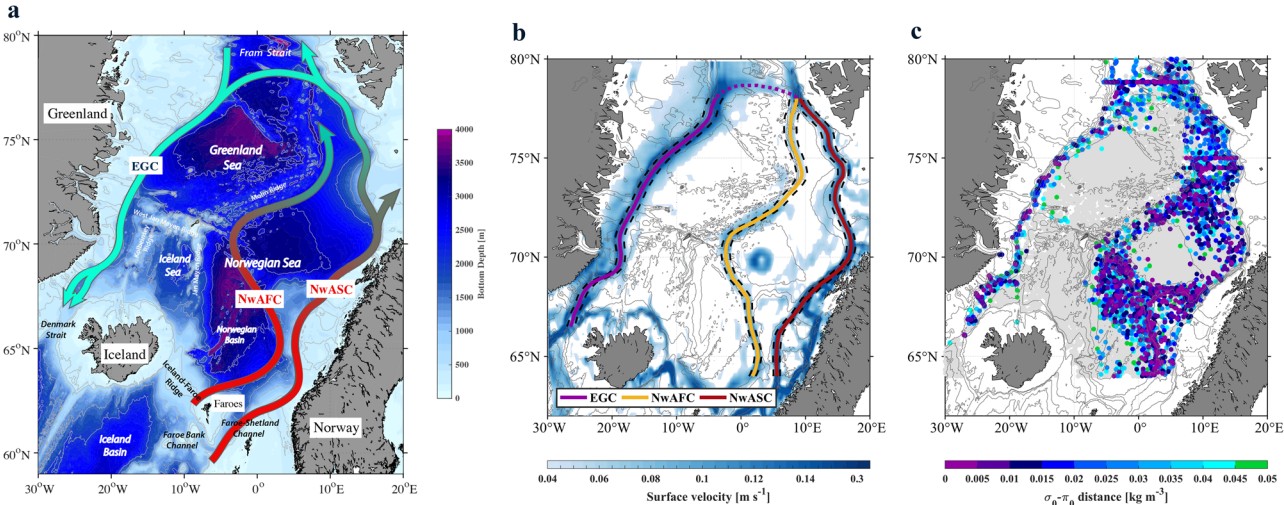

**Fig. 1 | Pathways of Atlantic Water along the rim current system in the Nordic Seas. a** Schematic circulation, modified from ref. [40]. The rim current system includes three boundary currents: the Norwegian Atlantic Slope Current (NwASC), the Norwegian Atlantic Front Current (NwAFC), and the East Greenland Current (EGC). **b** Composite of the long-term mean (2005–2018) sea surface velocity (absolute speed) from satellite observations and the GLORYS12 reanalysis product. The pathways of the three boundary currents in (**b**) were determined by maxima in the surface velocity field, shown by the solid lines. The black dashed lines denote

the boundary current region (distance to velocity maxima < = 25 km). The purple dashed line indicates the transition region that connects the eastern and western boundary currents. **c** Distribution of small $\sigma_0$-$\pi_0$ distance (where $\sigma_0$ and $\pi_0$ are the potential density and potential spicity, respectively, referenced to the sea surface) at 550–600 m, using historical hydrographic data from 2005 to 2018. Distances >0.05 kg m$^{-3}$ are shown by light-gray circles. The bathymetry (solid contours of 300, 500, 1000, 1500, 2000, 3000, 4000, and 5000 m) is from ETOPO1.

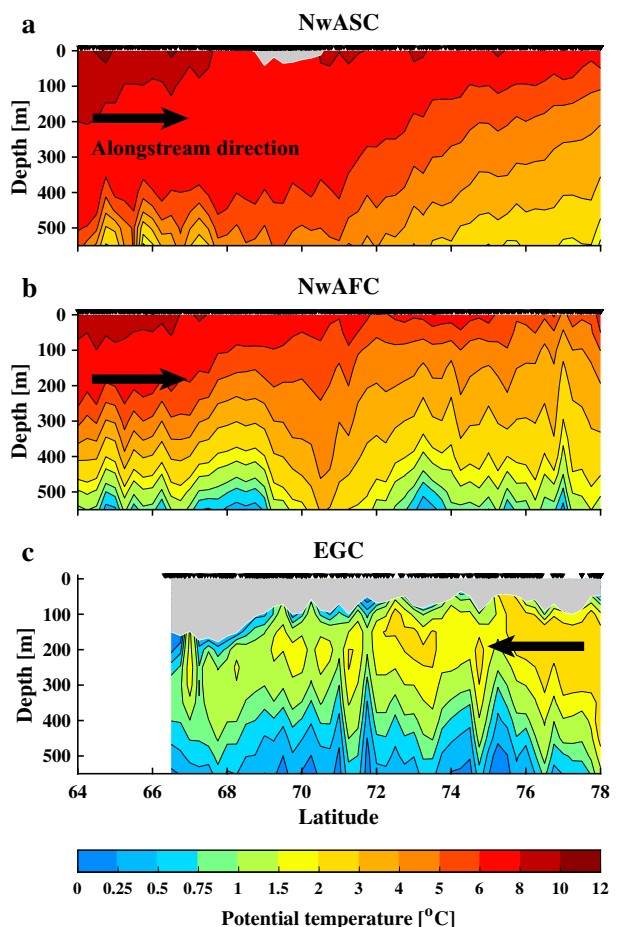

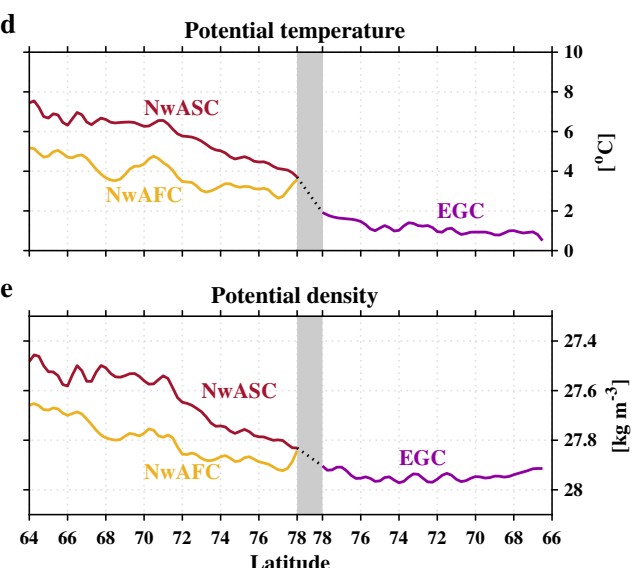

**Fig. 2 | Along-pathway transformation of Atlantic Water.** Vertical sections of temperature for the Atlantic Water layer (upper 550 m) along the three boundary currents: the (**a**) Norwegian Atlantic Slope Current (NwASC), **b** Norwegian Atlantic Front Current (NwAFC), and **c** East Greenland Current (EGC). The black arrow indicates the alongstream direction. The Polar Surface Water layer (defined by salinity <34.5 and potential density <27.7 kg m$^{-3}$) is masked by gray shading. The data points are indicated along the top of the plots. Depth-mean changes of (**d**) potential temperature and (**e**) potential density for the Atlantic Water layer along the NwASC (red), NwAFC (yellow), and EGC (purple). The transition portion that connects the two branches of the Norwegian Atlantic Current and the EGC is marked by the dotted line and gray bar.

Fig. 2d, e and Supplementary Fig. 2d). This is associated with the existence of the PSW layer at the surface of EGC, which seemingly prevents the ventilation of AW located below. However, as a result of sea ice retreat along east Greenland, together with shoreward advection of the PSW due to seasonally enhanced wind forcing, Våge et al.[17] argued that further ventilation of AW in the EGC can in fact occur in the late winter (their results were based on regional glider observations in two particular years). While the long-term mean results presented here show no evidence of such "re-ventilation" of AW, the section of Fig. 2c is likely biased by the large number of summertime hydrographic profiles; also, the re-ventilation process is a relatively new phenomenon. Furthermore, recent studies have demonstrated the importance of lateral processes on the water mass transformation in the EGC[13,14]. For example, the ocean response to a cold-air outbreak measured by a joint aircraft-ship campaign in winter 2018 was moderated by lateral advection and mixing near the EGC[14]. Hence, for the EGC, lateral processes seem to be equally important as air–sea fluxes in transforming the AW as it flows southward. To more robustly investigate the transformation of AW along the EGC, including its long-term variability, more observations along the pathway of the EGC are required (the EGC has been only sparsely observed compared with the two NwAC branches).

### Impact of air–sea heat loss on the cooling of Atlantic Water
We now seek to determine the impact of air–sea heat exchange on the transformation of AW along the two northward-flowing branches of the NwAC. Using high-resolution model simulations, recent studies have investigated the heat distribution and balance in the Nordic Seas[16,27,28], which has shed light on the mechanisms regulating the rim current transformation. However, uncertainty remains due to the problem of accurately representing the NwAFC in these models. Large differences in the alongstream velocity of the NwAFC (especially for the portion of current along the Mohn Ridge, see Fig. 2c in ref. [15], Fig. 5 in ref. [28], and Fig. 2 in ref. [26]) were found between the models and also between models and observations. This problem typically exists for models with limited assimilation of observational datasets.

We address this by applying the one-dimensional Price–Weller–Pinkel (PWP) mixed layer model in an advective framework following the NwASC and NwAFC, using observational hydrographic profiles as initial conditions where the two currents enter the domain from the south, subject to realistic atmospheric forcing from ERA5 along the pathways. The alongstream velocity is obtained from a reanalysis product (GLORYS12), which shows good agreement compared with direct historical velocity observations of the rim currents from ships, gliders, and moorings. Overall, the offset between GLORYS12 and observations is about 0.02 m s$^{-1}$ (see Supplementary Fig. 3). For each branch, we carried out 12 simulations, starting from the southern boundary with the mean hydrographic profile for each month of the year, and considered the ensemble average (interannual variability was not considered). Since PWP is one-dimensional and does not include lateral processes, this approach isolates the effect of air–sea cooling.

Differences between the simulated results and the observations are assumed to be due to lateral processes. The reader is referred to the "Methods" section for details.

The ensemble mean temperature change along the two branches of the NwAC from the monthly PWP simulations is presented in Fig. 3, with the standard error indicated. Comparing this with the long-term mean results from the hydrographic observations shows that the model well reproduces the net temperature change of AW along the NwAFC (Fig. 3a). The net cooling of AW along the NwAFC is −2.2 ± 0.2 °C in the observations (the error is the standard error from the linear fit) and −2.4 ± 0.6 °C in the model (the error was estimated by considering an uncertainty of 0.02 m s$^{-1}$ for the alongstream velocity of current). This indicates that the air–sea heat flux (including both turbulent and radiative components) accounts almost entirely for the net cooling of AW along this branch. We note that the model does not capture some of the local temperature variation, in particular the rapid cooling near 68°N and 73°N, and the rapid warming around 71°N. These are likely due to lateral interactions with the interior basins and gyres: the two regions of cooling due to mixing with the cold waters of the Norwegian Sea and Greenland Sea gyres, respectively, and the warming due to mixing with the warm waters of the Lofoten Basin. Such regionally lateral transfer of heat is consistent with the high-resolution numerical model results of ref. [16]. Our PWP model results indicate that the effect of these lateral processes balances out over the full pathway of the NwAFC.

By contrast, there is a pronounced difference between the model and the observations for the NwASC (see Fig. 3b and Supplementary Table 2), indicating that other processes, e.g., the lateral transfer heat, dominate the along-pathway temperature change of AW for this current. This is consistent with previous modeling studies that suggest a comparable or larger role of lateral eddy flux than the local air–sea heat loss on the cooling of AW[15], due to the large instability of the slope current[29]. Such a lateral transfer of heat could also be driven by the bifurcation of the slope current into the Barents Seas[30], and mixing with colder water in the interior of Norwegian Sea[12]. To summarize, our results reveal that air–sea heat fluxes play differing roles on the transformation of AW along its northern pathways in the Nordic Seas. This result is critical for understanding how the water mass transformation in the Nordic Seas will respond to the ongoing changes of atmospheric forcing, which is addressed in the next section.

## Long-term variability in the transformation of Atlantic Water and its relation to changes in air–sea heat fluxes

We now investigate the long-term change in the transformation of AW. This was done by applying a linear fit to the variation in temperature along the two pathways for each year from 1993 to 2018 to obtain the net cooling of AW along each branch (see "Methods" section and Supplementary Fig. 4 for details). Only the data in the upper 200 m were used, due to the fact that there is sparse yearly data coverage at deeper depths and that most of the temperature variability of AW are confined in the 0–200 m layer. The resulting long-term evolution of the net cooling of AW along the two NwAC branches is presented in the Fig. 4. The error bar indicates the standard error from the linear fit. Notably, a significant long-term reduction of 1.5 ± 0.6 °C from 1993 to

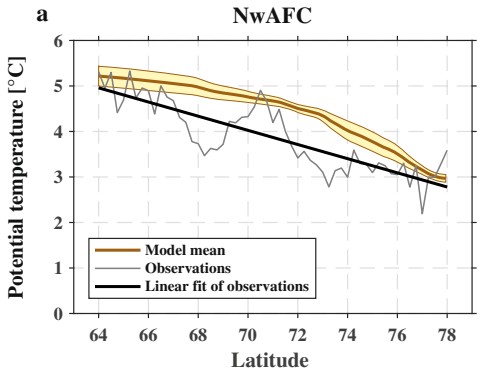
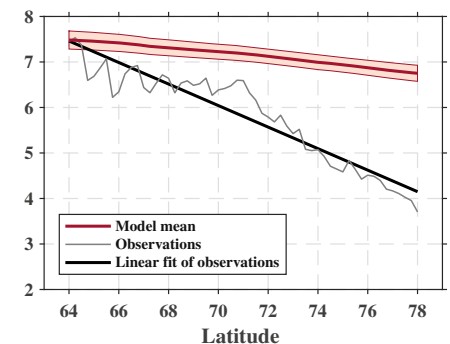

**Fig. 3 | Contribution of air–sea heat flux to the cooling of the Atlantic Water layer. a** Norwegian Atlantic Front Current (NwAFC) and **b** Norwegian Atlantic Slope Current (NwASC). The Atlantic Water layer is taken to be 0–550 m. The simulated temperature change (ensemble mean of monthly Price–Weller–Pinkel model runs) of Atlantic Water along the NwAFC and NwASC are shown in yellow and red, respectively, with the standard error shaded. The observed temperature change of Atlantic Water and a linear fit of this change are shown by the gray and black lines, respectively. The net cooling of Atlantic Water along the entire current is shown in Supplementary Table 2.

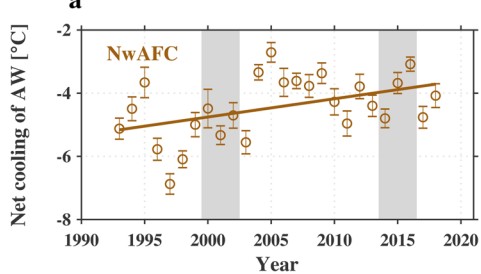
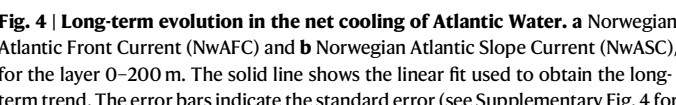
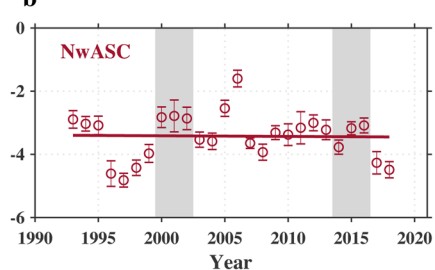

**Fig. 4 | Long-term evolution in the net cooling of Atlantic Water. a** Norwegian Atlantic Front Current (NwAFC) and **b** Norwegian Atlantic Slope Current (NwASC), for the layer 0–200 m. The solid line shows the linear fit used to obtain the long-term trend. The error bars indicate the standard error (see Supplementary Fig. 4 for the along-pathway cooling of Atlantic Water for each year). The gray shading indicates the two time periods considered in the text: 2000–2002 and 2014–2016 (see Supplementary Fig. 6 for the changes of air–sea turbulent heat flux, surface alongstream velocity, and sea surface height between these two time periods).

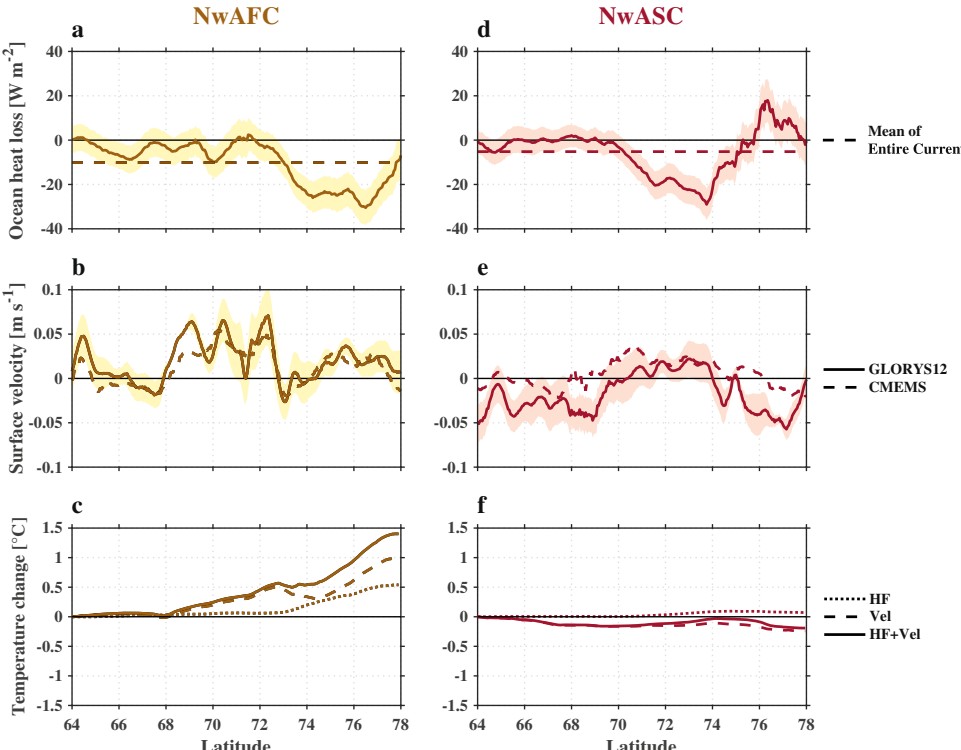

**Fig. 5 | Long-term changes in surface heat loss and alongstream velocity, and their impacts.** The left panels are the Norwegian Atlantic Front Current (NwAFC) and the right panels are the Norwegian Atlantic Slope Current (NwASC). The layer is 0–200 m. The solid lines in (**a**, **d**) show the long-term net changes of year-round mean air–sea turbulent heat flux along the two currents (from 1993 to 2018), with the standard error shaded. The dashed line denotes the mean change for the entire current. The solid and dashed lines in (**b**, **e**) show the long-term net change (from 1993 to 2018) of sea surface alongstream velocity from the reanalysis product (GLORYS12) and satellite observations (CMEMS), respectively. The short-dashed, long-dashed, and solid lines in (**c**, **f**) show the simulated temperature changes driven, respectively, by the long-term change of air–sea turbulent heat flux (**a**, **d**), by the long-term change of alongstream velocity (**b**, **e**), and by the combined effect of the two. The temperature change (in **c** and **f**) was obtained by removing the long-term mean result (control run) in the model (see text for details). A positive temperature change means a reduction in the cooling of the Atlantic Water.

2018 (*P* value in *t* test is 0.02) in the net cooling of AW occurred along the NwAFC. However, no such long-term trend was found for the NwASC (*P* value in *t* test > 0.1).

What are the reasons for this difference in long-term trends between the two branches? To answer this, we computed long-term changes of the year-round mean air–sea turbulent heat flux as well long-term changes in the alongstream velocity at the sea surface, as a function of along-pathway distance (Fig. 5a, b, d, e). A linear fit was used to obtain the net change of heat loss and alongstream velocity from 1993 to 2018 at each point along the two NwAC branches (using data within the 50 km swath corresponding to each long-term mean pathway). A sizeable reduction of ocean turbulent heat loss to the atmosphere was observed in the latitude range of 72–78°N for the NwAFC, and 70–76°N for the NwASC (Fig. 5a, d), with a maximum change of −30 ± 8 and −29 ± 7 W m⁻², respectively, for the two branches. Here we consider the turbulent components of the heat flux only because the long-term change of the radiative components is negligible (<2 W m⁻²) over the two NwAC branches. Regarding the alongstream velocity, different long-term trends were found between the NwAFC and NwASC: the velocity increased for nearly the entire NwAFC, while the velocity decreased for the northern and southern portions of the NwASC. Previous studies have suggested that a decrease of air–sea heat loss can lead to a reduction in the warm-to-cold transformation of water masses in the interior Nordic Seas[19]. At the same time, an increase of the alongstream velocity of the rim current has a similar effect due to the shorter residence time of water along the current (hence less time for it to experience cooling). We note that there is no significant long-term shift in the locations of the AW pathways defined by the maxima in the yearly surface velocity field

(see Supplementary Fig. 5a). While there is interannual variability in the pathway locations[31,32], the impact of this on the long-term changes of the net cooling of AW, air–sea turbulent heat flux, and alongstream velocity are small (see Supplementary Fig. 5b–g).

To better highlight these long-term changes, we focused on two selected time periods: an early period (2000–2002) and a late period (2014–2016). These two periods were chosen because the change in net cooling from early to late differs for the two NwAC branches (see the shading in Fig. 4) and both time periods are not associated with extreme NAO events. Consistent with the linear regression results, the air–sea turbulent heat loss along the two AW branches decreased from the early to the late period (compare Supplementary Fig. 6a and Fig. 5a, d). In terms of the alongstream velocity, the NwAFC strengthened over most of its path, while the NwASC weakened in the northern and southern parts of the domain (compare the vectors in Supplementary Fig. 6b and Fig. 5b, e). Notably, a gradient in the sea surface height (SSH) change between the two periods was found across the NwAFC, associated with the increase of alongstream velocity along this current (see Supplementary Fig. 6b). This is in line with the results of Broomé et al.[33] who documented an increasing trend of SSH in the eastern Nordic Seas from 1993 to 2017, dominated by the heat content change from the inflowing AW water. Raj et al.[34,35] suggested that the strength of NwAFC (NwASC) increases (decreases) during periods of negative NAO, when the cyclonic wind forcing over the Nordic Seas weakens. Based on the ERA5 reanalysis, there was no significant long-term change in the cyclonic wind forcing in the Nordic Seas. This implies that wind forcing cannot explain the long-term change in the velocity of the two currents and that it must be due to the asymmetric steric change in SSH. However, wind seemingly plays a role in regulating the

interannual variability (one can see the large interannual variability in the net cooling of AW along the two NwAC branches in Fig. 4). This requires further investigation.

To quantify the impact of the long-term changes of air−sea heat loss versus alongstream velocity on the cooling of AW along the two branches of the NwAC, the PWP model was again invoked. For each branch, four model runs were carried out with same initial conditions and differing forcing and current speeds. The first experiment is the control run where the model is forced by the long-term mean heat flux subject to the long-term mean alongstream velocity. The remaining three simulations corresponded to (a) the long-term change of turbulent heat flux alone; (b) the long-term change of alongstream velocity alone; and (c) the combination of the two. The differences in temperature of cases (a), (b), and (c) compared with the control run, as a function of along-pathway distance, are shown in Fig. 5c, f. For the NwAFC, both the decrease of ocean heat loss and the increase of the alongstream velocity was found to contribute to a significant reduction in the net cooling of the AW along the current. The combined effect results in a total reduction of 1.5 °C (Fig. 5c), which is remarkably consistent with the observed change in Fig. 4a (1.5 ± 0.6 °C from the linear fit).

By contrast, the temperature changes associated with the long-term changes of heat flux and alongstream velocity are much smaller for the NwASC (Fig. 5f), even though the magnitudes of the changes in air−sea forcing and velocity are comparable for the two currents. This is in line with the fact that no significant long-term trend was found for the net cooling of AW along this branch (Fig. 4b). A likely reason for this is that lateral processes, found to play a dominant role in the transformation of AW along the NwASC, dampen the sensitivity to long-term changes in air−sea heat loss and alongstream velocity. Our results thus reveal differing responses in the transformation of AW to long-term changes in air−sea heat flux and current strength, which should be taken into account when considering the response of the high-latitude AMOC to the changing climate.

## Discussion

Using historical hydrographic data, together with satellite data and a velocity reanalysis product, we investigated the role of air−sea heat flux on the transformation of Atlantic Water (AW) in the rim current system of the Nordic Seas. The densification of AW occurs primarily along the two branches of the Norwegian Atlantic Current: the Norwegian Atlantic Slope Current (NwASC) and the Norwegian Atlantic Front Current (NwAFC). In both cases, cooling in temperature dominates the change. The contribution of air−sea heat flux to this cooling was estimated by applying a one-dimensional mixing model following the flow. This demonstrated that air−sea heat flux accounts almost entirely for the cooling of AW along the NwAFC, while the lateral transfer of heat appears to dominate the temperature change of AW along the NwASC. A regression analysis of the along-pathway temperature change reveals a reduction in the net cooling of AW along the NwAFC since the early 1990s, due to a decrease in air−sea turbulent heat loss together with an increase in alongstream velocity. Such a long-term trend was not found for the NwASC, due to a dampened sensitivity of AW transformation to the changes in air−sea heat flux along this current.

The results presented in this study provide a Lagrangian view of how AW is transformed as it encircles the Nordic Seas. The use of a 1D mixed-layer model together with output from a reanalysis product following the current represents a new way to investigate the role of air−sea heat flux on the along-pathway cooling of AW. In addition to this Lagrangian approach, there are other alternative approaches that can be formally used to quantify the water mass transformation[36]. For example, in the traditional Eulerian framework, the impact of lateral processes on the local water mass transformation can be systematically quantified. Comparison of results from these different

approaches can improve our understanding of the rim transformation in the Nordic Seas. However, one should keep in mind that in future work it will be critical to faithfully simulate the NwAFC in high-resolution models. Our results imply that an inaccurate representation of this current can bias the sensitivity of the high-latitude AMOC to a changing climate.

Our major finding is that air−sea heat fluxes play differing roles on the transformation of AW along its northern pathways in the Nordic Seas. This provides insight into the link between atmospheric forcing and transformation of the upper limb of the AMOC, and how this might be impacted by ongoing changes in the high latitude North Atlantic climate. This is particularly important in light of the fact that a large contribution to the AMOC stems from the transformation occurring in the Nordic Seas[1]. It is important to note that the fate of the AW in the two NwAC branches is very different. While much of the transformed AW transported by the NwAFC returns southward via the EGC as part of the subpolar AMOC, a significant portion of the AW in the NwASC flows northward into the Arctic Ocean via the West Spitsbergen Current and to the Barents Sea via the North Cape Current. These latter two currents contribute to the overturning in the high Arctic and also play a fundamental role in the recently documented Arctic Atlantification[37,38]. Recent studies have demonstrated that the AW in the EGC, as well as that in the boundary current north of Svalbard in the Amundsen Basin, can be further transformed as a result of reduced ice cover in these regions[13,17,39]. Hence, it remains unclear how much the reduction in air−sea cooling along the NwAFC might be compensated for by the re-ventilation along the EGC. This makes it challenging to predict how the AMOC and Arctic overturning will respond to the differing effects of air−sea heat exchange on the two NwAC branches. Future observational and modeling studies are required to address this.

## Methods
### Hydrographic data
The hydrographic data used in this study, covering the time period 1993–2018, come from a recently compiled dataset for the Nordic Seas[40]. This dataset is a collection of various archives, including a dominant contribution from the Unified Database for Arctic and Sub-arctic Hydrography (UDASH)[41]. Additional details of each individual archive and the final combined dataset are provided in ref. [40].

### Velocity data
The 3-dimensional velocity data from 1993 to 2018 used in this study were obtained from the reanalysis product GLORYS12, https://resources.marine.copernicus.eu/product-detail/GLOBAL_MULTIYEAR_PHY_001_030/INFORMATION. This product is a global ocean eddy-resolving reanalysis covering the altimetry era (1993 onward). It is based on the NEMO platform, driven at the surface by European Center for Medium-Range Weather Forecasts (ECMWF) reanalysis (ERA-Interim and ERA5), and assimilating several observational datasets[42]. This product provides daily and monthly mean velocity data from the surface to the bottom on a standard grid with 1/12° lateral resolution, at 50 standard depth levels. Comparison with observations in the sub-polar North Atlantic have shown good agreement with respect to water mass properties, surface velocity, mesoscale activity, and sea-ice extent[43]. We assembled direct velocity observations of the boundary currents in the Nordic Sea from past shipboard measurements, moorings, and gliders, and compared these to the long-term mean GLORYS12 velocity (see Supplementary Fig. 3). The GLORYS12 product well represents the alongstream velocity for the three boundary currents, with a slight offset (about 0.02 m s⁻¹) likely due to the difference in the temporal resolution of the datasets. The low-frequency variability (interannual and decadal) has been proven to be well captured in the GLORYS12[42]. We also compared the long-term mean (2005–2018) sea surface velocity along the NwAFC from GLORYS12 (1/12°) with three

other reanalysis products: ARMOR3D (1/4°), ASTE (15–20 km), and GREP (1°) (see details of each product in Supplementary Table 1). This revealed that ARMOR3D seems to do a comparable job to GLORYS12 in capturing the current, but the other two products are deficient in this regard. In addition to the reanalysis product, satellite-based observations of sea surface height and sea surface velocity from 1993 to 2018 were used in this study (https://resources.marine.copernicus.eu/product-detail/SEALEVEL_GLO_PHY_L4_MY_008_047/INFORMATION), with daily temporal resolution and 1/4° spatial resolution. Note that the GLORYS12 velocity field was downscaled to a spatial resolution of 1/4° when making the composite sea surface velocity field shown in Fig. 1b. The composite was done by averaging the GLORYS12 and satellite data on the same grid.

### Air–sea heat flux data
The air–sea heat flux and wind stress data (1993–2018) used in this study were obtained from the ERA5 reanalysis from ECMWF, with 3-h temporal resolution and 1/4° spatial resolution (https://www.ecmwf.int/en/forecasts/datasets/reanalysis-datasets/era5). An evaluation of ERA5 air–sea turbulent heat flux and wind speed in the Nordic Seas shows good agreement with the observations, although the reanalysis data are less accurate over the marginal ice zone than over open water[44].

### $\sigma_0$-$\pi_0$ distance metric
A metric known as the $\sigma_0$-$\pi_0$ distance was applied to the hydrographic data to trace the pathways of AW in the Nordic Seas, where $\sigma_0$ is potential density and $\pi_0$ is potential spicity (referenced to the sea surface). Potential spicity is a thermodynamic variable whose contours are orthogonal to density contours in potential temperature–salinity ($\theta$-S) space[45]. This metric has been successfully applied to obtain the pathways of the densest overflow water that emanates from the Greenland Sea gyre[40]. Since $\sigma_0$ and $\pi_0$ are orthogonal to each other, the distance computed in $\sigma_0$-$\pi_0$ space is more meaningful than that in $\theta$-S space. Based on the dense Atlantic Water mode ($\sigma_{0,1} = 28.0$ kg m$^{-3}$, $\pi_{0,1} = -2.98$ kg m$^{-3}$) identified in Supplementary Fig. 1, the $\sigma_0$-$\pi_0$ distance between any water parcel ($\sigma_{0,2}$, $\pi_{0,2}$) in the Nordic Seas and this dense mode is calculated by the following equation:

$$D_{1,2} = \sqrt{(\sigma_{0,1} - \sigma_{0,2})^2 + (\pi_{0,1} - \pi_{0,2})^2} \qquad (1)$$

The smaller the $\sigma_0$-$\pi_0$ distance ($D_{1,2}$), the more similar the water parcel is to the dense Atlantic Water mode. In addition to our defined dense AW mode, other less dense modes of AW were considered by choosing points along the ridge of maximum occurrence percentages in Supplementary Fig. 1 (the black triangles). However, the resulting $\sigma_0$-$\pi_0$ distance maps for these modes did not resemble the circulation map of Fig. 1a. This is because these modes correspond to shallower waters (<550 m) whose properties are significantly modified as they flow northward.

### One-dimensional mixing model following the current
The Price–Weller–Pinkel (PWP) mixed layer model[46] was used to simulate the temperature change of AW along the two branches of NwAC. This is done via the following steps. First, the monthly mean profiles (0–550 m) of temperature, salinity and density at the southern end of the pathways (64–64.5°N) were calculated and used as the initial conditions in the model. Then, for each month's initial profile, the model is forced by the timeseries of air–sea heat flux and wind stress from ERA5, where these were determined by the time and location of the profile as it advects along the pathway according to the depth-mean (0–550 m) alongstream velocity of the current from the GLORYS12 product. For each flow branch we carried out 12 simulations corresponding to the 12 months of the year. The surface freshwater fluxes are excluded because their impact on the

temperature change of AW layer is negligible. The total time for a profile to propagate along the entire pathway is approximately 6 (20) months for the NwASC (NwAFC). Since the PWP is one-dimensional and does not include lateral processes, the differences between the PWP results and observations are assumed to be due to lateral processes (the vertical flux of heat at the bottom of AW layer is negligible because the maximum mixed layer depth along the two NwAC branches is shallower than 550 m). We note that the PWP model has a limitation in estimating the air–sea cooling of AW along the EGC. This is due to the thick and fresh Polar Surface Water layer in the EGC. The existence of this layer results in a quick formation of sea ice in the model, which prevents air–sea cooling of the underlying AW.

### Composite vertical sections
A Laplacian–spline interpolation method[47] was applied to the historical hydrographic profiles within a 50 km swath centered on the NwAC pathways to construct composite vertical sections of temperature, salinity, and density along the pathway. The final gridded sections have a vertical resolution of 25 m and a horizontal resolution of 0.25°.

### Linear fits
A linear regression fit ($y = a \cdot x + b$) was used to obtain the net cooling of AW along the NwASC and NwAFC (see Fig. 3 and Supplementary Fig. 4), where $x$ is the latitude, $y$ is the temperature change along the current, and $a \cdot \Delta x$ ($\Delta x = 78°N–64°N$) is the net cooling of AW. Another linear fit ($y = a \cdot t + b$) was applied to obtain the long-term net change of air–sea heat flux and sea surface velocity at each point along the two pathways, where $t$ is the year, $y$ is the year-long mean turbulent heat flux and sea surface velocity, and $a \cdot \Delta t$ ($\Delta t = 2018–1993$) is the net change of air–sea turbulent heat flux and sea surface velocity from 1993 to 2018. The standard error of estimated coefficients and the $P$ value in the Student's $t$ test of the linear fit are provided.

## Data availability
The UDASH Hydrographic data are available at https://doi.pangaea.de/10.1594/PANGAEA.872931. The GLORYS12 reanalysis data are available at https://data.marine.copernicus.eu/product/GLOBAL_MULTIYEAR_PHY_001_030/description. The satellite-based sea surface height and sea surface velocity data are available at https://data.marine.copernicus.eu/product/SEALEVEL_GLO_PHY_L4_MY_008_047/description. The ERA5 air–sea heat flux and wind stress data are available at https://www.ecmwf.int/en/forecasts/datasets/reanalysis-datasets/era5. Details of each individual data source used in this study are provided in Supplementary Table 1.

## Code availability
The source code of PWP model is available at http://www.po.gso.uri.edu/rafos/research/pwp/. The code to reproduce the figures is available upon request from the corresponding author.

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

## Acknowledgements

We thank Ailin Brakstad, Kjell Arild Orvik, Kjetil Våge, and Ilker Fer for providing the historical hydrographic, mooring, and glider data used in this study. Funding for the study was provided by the US National Science Foundation under grants OCE-1756361 and OCE-1948505 (J.H. and R.P.).

## Author contributions

J.H., R.P., and Z.C. assembled and analyzed the data; J.H. and R.P. wrote the paper; J.H., R.P., Z.C., and R.H. interpreted the results and clarified the implications.

## Competing interests

The authors declare no competing interests.
