## [Peer Review File · Nature Communications]

Role of air-sea heat flux on the transformation of Atlantic Water encircling the Nordic SeasREVIEWER COMMENTS

Reviewer #1 (Remarks to the Author):

Using observations and models, the authors diagnose warm-to-cold transformation of Atlantic Water (AW) in the Nordic Seas. Two distinct pathways are emphasized: the coastal Slope Current and an interior 'Front Current'. In the Slope Current, AW cooling is a consequence of lateral mixing processes, while AW cooling along the interior pathway is primarily attributed to surface heat loss. Based on this process-level knowledge, changes over 1993-2018 are investigated. Reduced heat loss along the interior pathway contrasts with no significant change along the coastal pathway. Data and Methods are well documented, sufficient for reproduction of the analyses. The findings are clearly presented and should be of interest to ocean and climate scientists concerned with the North Atlantic circulation.

This being said, the diagnostic approaches used here are highly bespoke: describing temperature changes for target AW layers and pathways, based on a water mass metric; inferring the role of heat fluxes (surface or interior) with an 'offline' 1D model advected along identified pathways. Water mass transformation rates (attributed to surface fluxes or mixing) may be otherwise formally quantified (see Groeskamp et al., 2019), while property change from a Lagrangian perspective may be quantified with data from model hindcasts such as GLORYS12 by calculating large ensembles of particle trajectories (see Van Sebille et al., 2018) that record property changes along selected pathways. Such alternative approaches could be considered by introduction or discussion, and the bespoke approaches perhaps thus justified.

I have three Specific Comments:

- 1. Please provide a brief definition of 'potential spicity', likely unfamiliar to most readers.**
- 2. Over 1993-2018, velocity decreases (increases) for the NASC (NAFC) are attributed to the long-term reduction of cyclonic wind forcing over the Nordic Seas, citing Roshin et al. (2018, 2019), but these references are not listed; it would further be helpful for this wind-forced change in circulation to be briefly explained.**
- 3. As well as relevance for the AMOC, it seems that changes in AW modification are consequential for the Arctic; over the study period, attention has been drawn to 'Atlantification' of the Barents Sea (see Lind et al. 2018); it would be appropriate to mention this in 'Conclusion and discussion'. Perhaps it is further useful to note that the NASC conveys much of the AW inflow to the Arctic via the West Spitzbergen Current and to the Barents Sea via the North Cape Current (contributing to Arctic Atlantification), while much of the NAFC feeds the offshore EGC (potentially impacting the AMOC).**

References:

Groeskamp, S., Griffies, S. M., Iudicone, D., Marsh, R., Nurser, A J.G., and J.D. Zika (2019). The water mass transformation framework for ocean physics and biogeochemistry. Annual Review of Marine Science, 11(1), 271-305. <https://doi.org/10.1146/annurev-marine-010318-095421>

Lind, S., Ingvaldsen, R.B., and T. Furevik (2018). Arctic warming hotspot in the northern Barents Sea linked to declining sea-ice import, Nature Climate Change, 8, 634-639

Van Sebille, E., et al. (2018). Lagrangian ocean analysis: fundamentals and practices. Ocean Modelling, 121, 49-75. <https://doi.org/10.1016/j.ocemod.2017.11.008>

Reviewer: Professor Robert Marsh

Reviewer #2 (Remarks to the Author):

Review of Huang et al. "Role of air-sea heat flux on the transformation of Atlantic Water as it encircles the Nordic Seas".

The paper by Huang et al investigates the transformation of Atlantic Water (AW) as it encircles the Nordic Seas. This densification of AW contributes to the production of Nordic Seas overflow waters, which are an important component of the lower limb of the AMOC. Using hydrographic observations, reanalysis velocities, and a one-dimensional mixing model, the authors find that air-sea heat fluxes play different roles in the transformation of AW along different branches. As a consequence, different branches of AW show different long-term cooling trends.

In general, I find the paper interesting and well-written. Some of the results in the paper I believe to be fairly well-established in the literature (the mean pathways and modification, Sections 2-3; I will elaborate on this below), but the results are presented in a nice and coherent way using extensive data and interesting methods. The authors acknowledge that previous studies have investigated water mass transformation along the AW pathway, but state that (l.146-147) "results remain unclear due to the large differences between models and also between models and observations". These large differences are, however, not explained and it is not clear how the presented analysis is able to solve these (apparent) disagreements. The conclusion that lateral transfer of heat is important for the slope current is also not new. For example, Isachsen et al (2012) found that "cooling of the current by lateral eddy fluxes is comparable to or larger than the local heat loss to the atmosphere". Hence, although I find the results interesting to specialists in the field, I believe the novelty of the analysis and the main findings should be better highlighted. In my opinion, the main new finding is the different long-term changes in the two AW branches, but this difference is not much explored (as acknowledged on l.261).

Further general comments:

1) As mentioned above, I believe that the pathways and transformation of AW within the Nordic Seas are well established (papers cited in the manuscript, as well as e.g., Furevik 2001; Orvik & Niiler 2002). I understand of course that the AW pathways need to be defined for the specific dataset used here, but in my opinion too much time/space is spent on establishing this. For example, given that you end up using the maximum surface velocity for the analysis of along-path water mass transformation, the sigma-pi analysis seems somewhat redundant (although a nice test of robustness which could be added to Methods/Supplementary).

2) The roles of air-sea heat flux and lateral mixing (a key conclusion in the paper) should be better quantified. I find the description of Figure 3 to be quite qualitative. From l.41-42, I expected that the relative contribution of these two processes would be quantified.

Also, the conclusion is that air-sea heat fluxes accounts almost entirely for the net cooling along the NAFC. At the same time the authors find that lateral heat transfer dominates much of the along-path variations, but that these balances out over the pathway (l.164-173). The only time the observed and modeled temperatures agree is at the start and end of the pathway. Considering the linear fit, there are also large discrepancies. So is it fair then to say that air-sea heat fluxes dominate?

3) The temporal variability in AW cooling (Figures 3 and 4) is an interesting finding. However, I think more should be done to try to explain these changes and their drivers. It is mentioned (l.207-209) that differing long-term trends in alongstream velocity can be explained by the wind field over the Nordic Seas, which would strengthen (weaken) the NAFC (NASC). However, in the region with the largest strengthening of the NAFC (70-74N) the NASC also strengthens. What can explain this? And more generally, what

is the reason for the large differences in velocity trends along the different pathways? Relatedly, it wasn't clear to me whether temporal variations in the AW pathways (e.g., Richter & Maus 2011) are taken into account (besides using data within 25 km of the pathways). As you are using the surface velocity field from GLORYS12 to define the AW pathways, it would be interesting to see to what extent the results change if you consider monthly or annual variations in pathways.

Lastly, are there any long-term changes for the EGC? As you mention in the manuscript, this is a region which has seen large changes recently (e.g., Moore et al. 2022). It is therefore a bit surprising that changes in the EGC are not considered.

4) I miss a short discussion on the data and methods used in the paper. For example, are there any limitations associated with the 1D mixed-layer model (see also comment to l.158-159 below)? For the EGC, you mention that observations are biased toward summer (l.130). How is this for the northern AW branches? Any uncertainties with the atmospheric forcing (see comment below to l.294)?

Detailed comments:

l.30. I suggest you add a few more references here. Also, Chafik et al. (2021) is more of an opinion-paper published in EOS.

l.36-42. As mentioned, I think the authors should better establish the novelty of their analysis. Papers on the importance of lateral heat transfer from the NASC exist before 2021 (e.g., Isachsen et al. 2012). It is not clear to me why the authors choose to highlight an unpublished paper from 2022 (Renfrew et al; l.38-42).

l.48. The results from Smedsrud et al. (2022) show no significant changes in heat loss over the Nordic Seas. The decreased heat loss only takes place over the Greenland and Iceland Seas. Please comment.

l.49. "profound impacts" – please elaborate.

l.54. "large-scale ramifications" – please elaborate.

l.58. The Norwegian Atlantic Current is normally abbreviated NwAC to separate it from the North Atlantic Current (NAC).

l.96. It is not clear to me (from Methods) how the satellite observations and reanalysis product was combined (figure caption says "composite").

l.128-131. Given these recent changes in AW ventilation, are there any changes in AW cooling along the EGC?

l.133-138. Why don't you assess the roles of air-sea heat fluxes and lateral transport in the same way as for the other two branches? Basing this conclusion on measurements from one year is not convincing.

l. 146-147: Please elaborate on these large differences and how your approach is different (and better, noting that you also make use of a model/reanalysis and a mixed-layer model).

l.158-159: "Differences between the simulated results and observations are assumed to be due to lateral processes." How valid is this assumption? Could the differences be due to something else? Please elaborate.

l.165. Figure 3b is presented before 3a. Reverse order?

l.191. Do conclusions depend on the chosen depth averaging? Would you expect results to change if you used a time-dependent mixed-layer depth? Also, earlier in the manuscript you used 550 m (Figure 2 and 3). Do these figures change for 0-200m?

l.194: "significant long-term reduction". From Figure 4 it looks like there minimum AW cooling took place in the early 2000s after which it has increased. A similar development can be seen for temperature in the Norwegian Sea (Orvik 2022). I understand (and agree) that a linear trend is used in the analysis, but think the authors should make it clear in the text whether they interpret this as a long-term trend or (decadal) variability.

l.233-236. Wouldn't lateral heat transfer depend somewhat on the alongstream velocity?

l.236-238/l.258. Could the authors speculate/discuss how the identified difference in water mass transformation along the different AW pathways could impact the response of the high-latitude AMOC to a changing climate? Would this make the AMOC more or less sensitive to change? And would this imply that coarse resolution climate models

that don't resolve the NAFC are more/less sensitive to climate change?

I.243. "The densification of AW was found to occur..." This phrasing makes it sound like this is a new finding of this paper. Suggest rephrasing.

I.260-265. I suggest that the authors rather highlight the new findings (novelty) of this study rather than ending with all that remains to be understood.

I.294. Are your results sensitive to the chosen reanalysis product? Temporal variability can be quite different for different products (e.g., Carton et al. 2011).

References

Carton et al. (2011). doi:10.1029/2011JC007102

Furevik (2001). [https://doi.org/10.1016/S0967-0637\(00\)00050-9](https://doi.org/10.1016/S0967-0637(00)00050-9)

Isachsen et al. (2012). doi:10.1029/2012JC007935

Orvik & Niiler (2002). doi:10.1029/2002GL015002

Orvik (2022). <https://doi.org/10.1029/2021GL096427>

Richter & Maus (2011). <https://doi.org/10.1029/2011JC007311>

Reviewer #3 (Remarks to the Author):

Review of Huang et al 2022

The paper is clearly written, logically presented, and have important results. It suggests that the cooling of the NAFC is reduced over the period 1993-2018. In contrast, changes in cooling are less clear in the NASC where advection play a bigger role. Before possible publication there are some major points that the authors should address. This is especially related to the validity of using a time-constant pathways that for the NAFC branch that may lead to methodological problems. If this can be sorted out convincingly this paper would be a valuable contribution with both regional and larger scale interest.

Major points

1. Fig1b now basically show the extent of the Atlantic layer but avoiding the Lofoten Basin since the lower interface of the Atlantic Layer here is deeper, typically at 800m. Using the 550-600 m layer for defining the pathways, minimum in density-spiciness space, is not justified. Please explain.

2. The hydrographic database used here is from 1993-2018. Why do limit to the period 2005-2018 when defining the pathways, sigma-spiciness distance. Would they look different during the early period? The reason for bringing this up is that while the slope current is basically trapped by the shelf break, the NAFC may have larger potential for zonal shift (see e.g Blindheim et al., 2000), that also may vary meridionally. You assume a fixed pathway in this study. However, can you rule out that your reduced cooling in the NAFC is not a result of varying zonal extent of the Atlantic Water? Figure 4 suggest that the reason for the different cooling trends in the Front and Slope currents are basically explained by the "cold" water in the northern part of the NAFC in the early part (before 2004). The authors need to provide convincing arguments that the conclusion is not a result of a unrealistic fixed pathway for the NAFC, since it is in the core of the main result and conclusion of this paper.

Minor points

1. References: Roshin 2018,2019 should be Raj;

2. Fig 1c, difficult to see if there is a maximum in velocity (speed), especially for the NAFC but also partially for the NASC.

3. Please provide the argument why reduced cyclonic wind forcing should increase the NAFC.

We appreciate the reviewers' careful and constructive comments on the manuscript. Our responses are listed in blue.

Reviewer #1 (Remarks to the Author):

Using observations and models, the authors diagnose warm-to-cold transformation of Atlantic Water (AW) in the Nordic Seas. Two distinct pathways are emphasized: the coastal Slope Current and an interior 'Front Current'. In the Slope Current, AW cooling is a consequence of lateral mixing processes, while AW cooling along the interior pathway is primarily attributed to surface heat loss. Based on this process-level knowledge, changes over 1993-2018 are investigated. Reduced heat loss along the interior pathway contrasts with no significant change along the coastal pathway. Data and Methods are well documented, sufficient for reproduction of the analyses. The findings are clearly presented and should be of interest to ocean and climate scientists concerned with the North Atlantic circulation.

This being said, the diagnostic approaches used here are highly bespoke: describing temperature changes for target AW layers and pathways, based on a water mass metric; inferring the role of heat fluxes (surface or interior) with an 'offline' 1D model advected along identified pathways. Water mass transformation rates (attributed to surface fluxes or mixing) may be otherwise formally quantified (see Groeskamp et al., 2019), while property change from a Lagrangian perspective may be quantified with data from model hindcasts such as GLORYS12 by calculating large ensembles of particle trajectories (see Van Sebille et al., 2018) that record property changes along selected pathways. Such alternative approaches could be considered by introduction or discussion, and the bespoke approaches perhaps thus justified.

We thank the reviewer for the comments and suggestions. In the revised manuscript, we now discuss other alternative approaches that could be formally used to study the water mass transformation.

I have three Specific Comments:

1. Please provide a brief definition of ‘potential spicity’, likely unfamiliar to most readers.

Potential spicity is a thermodynamic variable representing density-compensating temperature and salinity variance. Hence, the contours of potential spicity are orthogonal to the contours of potential density when plotted in potential temperature–salinity (θ -S) space. A brief definition of ‘potential spicity’ has been provided in the Methods section.

2. Over 1993-2018, velocity decreases (increases) for the NASC (NAFC) are attributed to the long-term reduction of cyclonic wind forcing over the Nordic Seas, citing Roshin et al. (2018, 2019), but these references are not listed; it would further be helpful for this wind-forced change in circulation to be briefly explained.

Thanks for this comment. The references Roshin et al. (2018, 2019) have been corrected to Raj et al. (2018, 2019), sorry for the omission. Raj et al. (2018, 2019) suggested that the strength of NwASC (NwAFC) decreases (increases) during the negative NAO events, when the cyclonic wind forcing over the Nordic Seas weakens. Based on your comment we have considered this more carefully. Close inspection of the ERA5 reanalysis reveals only a slight weakening of cyclonic wind forcing in the Greenland Sea, and not over the eastern Nordic Seas. This implies that there should be other mechanisms driving the long-term change of circulation of the two NwAC branches. To better understand this, in the revised manuscript we compared the changes of sea surface height (SSH) between two time periods: 2000-2002 and 2014-2016. We note that both time periods are not associated with extreme NAO events, but there are significant changes with respect to net cooling of AW, air-sea heat flux, and surface velocity along NwAFC (see revised Figure 4 and Supplementary Figure 6). The SSH change map provides an explanation for the long-term change in the circulation along NwAFC. In particular, the increase of SSH is stronger in the

eastern Nordic Seas, especially in the Lofoten Basin. This change leads to an SSH gradient across the NwAFC which intensifies the strength of current. Such a long-term change in the SSH field has also been found in Broomé et al. (2020), who suggested that this change was dominated by the long-term heat content change in the Nordic Seas due to warming of the inflowing AW water. In summary, while the wind forcing can play a role in the circulation change during the extreme NAO years (i.e., interannual variability), the long-term change in the velocity field seems to be associated with the asymmetric heat content change in the Nordic Seas. We have explained all of this in the revision.

3. As well as relevance for the AMOC, it seems that changes in AW modification are consequential for the Arctic; over the study period, attention has been drawn to ‘Atlantification’ of the Barents Sea (see Lind et al. 2018); it would be appropriate to mention this in ‘Conclusion and discussion’. Perhaps it is further useful to note that the NASC conveys much of the AW inflow to the Arctic via the West Spitzbergen Current and to the Barents Sea via the North Cape Current (contributing to Arctic Atlantification), while much of the NAFC feeds the offshore EGC (potentially impacting the AMOC).

We now include this in the “conclusion and discussion” section. We also note that the NwASC participates in the overturning of the high Arctic.

Reviewer #2 (Remarks to the Author):

The paper by Huang et al investigates the transformation of Atlantic Water (AW) as it encircles the Nordic Seas. This densification of AW contributes to the production of Nordic Seas overflow waters, which are an important component of the lower limb of the AMOC. Using hydrographic observations, reanalysis velocities, and a one-dimensional mixing model, the authors find that air-sea heat fluxes play different roles in the transformation of AW along different branches. As a consequence, different branches of AW show different long-term cooling trends.

In general, I find the paper interesting and well-written. Some of the results in the paper I believe to be fairly well-established in the literature (the mean pathways and modification, Sections 2-3; I will elaborate on this below), but the results are presented in a nice and coherent way using extensive data and interesting methods. The authors acknowledge that previous studies have investigated water mass transformation along the AW pathway, but state that (1.146-147) “results remain unclear due to the large differences between models and also between models and observations”. These large differences are, however, not explained and it is not clear how the presented analysis is able to solve these (apparent) disagreements. The conclusion that lateral transfer of heat is important for the slope current is also not new. For example, Isachsen et al (2012) found that “cooling of the current by lateral eddy fluxes is comparable to or larger than the local heat loss to the atmosphere”. Hence, although I find the results interesting to specialists in the field, I believe the novelty of the analysis and the main findings should be better highlighted. In my opinion, the main new finding is the different long-term changes in the two AW branches, but this difference is not much explored (as acknowledged on 1.261).

Thank you for this input. In the revised manuscript, we have highlighted the novelty and the main findings of our study in the “conclusion and discussion” section. One of the novel aspects of our study is the Lagrangian approach used to isolate the role of air-sea heat flux. Reviewing the

previous high-resolution model studies (e.g., Isachsen et al. 2012; Ypma et al., 2020; Spall et al., 2021; Treguier et al., 2021), we found that the NwAFC was not well resolved in these models. In particular, the strength of NwAFC varies between models and also between models and observations. The GLORYS12 reanalysis assimilates several observational datasets and provides an accurate estimate of the velocity of the NwAFC and NwASC (as demonstrated in our manuscript). Therefore, we can combine this product with a 1D mixed-layer model to robustly isolate the effects of air-sea cooling.

Perhaps our most important finding is that air-sea heat flux plays differing roles on the net cooling along the two AW branches. This conclusion has not been previously established due to the fact that earlier studies (e.g., Isachsen et al. 2012) mainly focused on NwASC, while the transformation of AW along the NwAFC has not been considered due to the difficulty in accurately reproducing the NwAFC in numerical models. Our finding is essential for understanding the different long-term changes in the two AW branches. This is further explored in the revised manuscript (see details in the text and our responses below).

Further general comments:

1) As mentioned above, I believe that the pathways and transformation of AW within the Nordic Seas are well established (papers cited in the manuscript, as well as e.g., Furevik 2001; Orvik & Niiler 2002). I understand of course that the AW pathways need to be defined for the specific dataset used here, but in my opinion too much time/space is spent on establishing this. For example, given that you end up using the maximum surface velocity for the analysis of along-path water mass transformation, the sigma-pi analysis seems somewhat redundant (although a nice test of robustness which could be added to Methods/Supplementary).

Point taken. In the revised manuscript we have reorganized the Results section 1. Some details of sigma-pi analysis have been moved to the Methods section.

2) The roles of air-sea heat flux and lateral mixing (a key conclusion in the paper) should be better quantified. I find the description of Figure 3 to be quite qualitative. From 1.41-42, I expected that the relative contribution of these two processes would be quantified.

To completely quantify the relative roles of air-sea fluxes and lateral processes on the cooling of the AW, a heat-balanced 3D high-resolution model is required. However, by considering previous high-resolution model studies (e.g., Isachsen et al. 2012; Ypma et al., 2020; Spall et al., 2021; Treguier et al., 2021), we found that the NwAFC was not well resolved in these models. Hence it is impossible to use such a model to quantify the roles of the different processes on the water mass transformation along NwAFC. This motivated us to use an ocean state estimate that is ground-truthed by in-situ data, together the 1D mixed-layer model. The model effectively isolates the contribution of air-sea heat flux to the net cooling of AW, hence the quantification of the contribution of air-sea heat flux is indeed robust. The remaining contribution, which is dominated by lateral processes, can be estimated as the difference between the model and observations. We note that Isachsen et al (2012) argued that cooling of the NwASC by lateral eddy fluxes is comparable or greater than the local air-sea heat loss. This is consistent with our 1D model result. In the revised manuscript, we have added a Supplementary Table showing the contribution of air-sea cooling and lateral cooling to the observed net cooling of AW water, with uncertainties included.

Also, the conclusion is that air-sea heat fluxes accounts almost entirely for the net cooling along the NAFC. At the same time the authors find that lateral heat transfer dominates much of the along-path variations, but that these balances out over the pathway (1.164-173). The only time the observed and modeled temperatures agree is at the start and end of the pathway. Considering the

linear fit, there are also large discrepancies. So is it fair then to say that air-sea heat fluxes dominate?

In the paper we focus on the net change over the entire current. We acknowledge that the model does not capture some of the local lateral processes, and we explain that the effect of these regional lateral heat exchanges balances out over the full pathway. That said, the net change in temperature over the full path length of the current is due to air-sea cooling, as we demonstrate. In particular, observationally calculated net cooling along the NwAFC is -2.2 ± 0.2 °C, while the model-based value is -2.4 ± 0.6 °C. This information is now included as part of Supplementary Table 2.

3) The temporal variability in AW cooling (Figures 3 and 4) is an interesting finding. However, I think more should be done to try to explain these changes and their drivers. It is mentioned (1.207-209) that differing long-term trends in alongstream velocity can be explained by the wind field over the Nordic Seas, which would strengthen (weaken) the NAFC (NASC). However, in the region with the largest strengthening of the NAFC (70-74N) the NASC also strengthens. What can explain this? And more generally, what is the reason for the large differences in velocity trends along the different pathways? Relatedly, it wasn't clear to me whether temporal variations in the AW pathways (e.g., Richter & Maus 2011) are taken into account (besides using data within 25 km of the pathways). As you are using the surface velocity field from GLORYS12 to define the AW pathways, it would be interesting to see to what extent the results change if you consider monthly or annual variations in pathways.

Thanks for these comments. In the revised manuscript, we have provided more explanation about the long-term change in the velocity of the two NwAC branches. Raj et al. (2018, 2019) suggested that the strength of NwAFC (NwASC) increases (decreases) during negative NAO periods, when the cyclonic wind forcing over the Nordic Seas weakens. This is the reason we felt that the long-term changes for the alongstream velocity of two NAC branches were likely associated with the

changes of wind forcing. However, we have now carefully analyzed the ERA5 reanalysis data which revealed only a slight long-term weakening of cyclonic wind forcing in the Greenland Sea, and no such trend in the eastern Nordic Seas. This implies that wind forcing is not driving the long-term change of circulation for the two NwAC branches.

To better understand this, we compared the changes of sea surface height (SSH) between two time periods: 2000-2002 and 2014-2016. We note that both time periods are not associated with extreme NAO events, but there are significant changes with respect to net cooling of AW, air-sea heat flux, and surface velocity along NwAFC (see revised Figure 4 and the new Supplementary Figure 6). The SSH change map provides an explanation for the long-term change in the circulation along NwAFC. In particular, the increase of SSH is stronger in the eastern Nordic Seas, especially in the Lofoten Basin. This change leads to an SSH gradient across the NwAFC which intensifies the strength of current. Such a long-term change in the SSH field has also been found in Broomé et al. (2020), who suggested that this change was dominated by the long-term heat content change in the Nordic Seas due to warming of the inflowing AW water. In summary, while the wind forcing can play a role in the circulation change during the extreme NAO years (i.e., interannual variability), the long-term change in the velocity field seems to be associated with the asymmetric heat content change in the Nordic Seas. We have explained all of this in the revision.

Regarding the question about temporal variations in the AW pathways, we agree that monthly or yearly changes in the pathway could impact the transformation of AW in a particular month or year. However, we do not think that such variability will impact the long-term changes presented in this study. The new Supplementary Figure 5a shows the yearly pathways of two NwAC branches defined by the maxima in the surface velocity field in each year. While one can see that the pathways of NwAFC are more variable than that of NwASC, there is no clear evidence of a long-term change in pathway for either current. Since the width of each current is defined to be a 50 km swath, our results concerning the long-term changes of net cooling of AW, air-sea heat flux, and

along-stream velocity are not sensitive to the yearly variation of the pathways (see Supplementary Figure 5b-g). This is explained in the revision.

Lastly, are there any long-term changes for the EGC? As you mention in the manuscript, this is a region which has seen large changes recently (e.g., Moore et al. 2022). It is therefore a bit surprising that changes in the EGC are not considered.

1.128-131. Given these recent changes in AW ventilation, are there any changes in AW cooling along the EGC?

As stated in the original manuscript, the current observational data coverage is not sufficient to resolve the year-to-year changes of water mass transformation along the EGC. While studies have reported on long-term changes of air-sea heat flux and sea ice near the EGC (e.g., Moore et al. 2022), it remains unclear how these changes impact the transformation of AW along EGC. Våge et al. (2018) suggested re-ventilation of AW along the EGC, but their results were based on targeted observations from two particular years. In the revised manuscript, we explain more explicitly this limitation for investigating the long-term changes in the transformation of AW along the EGC.

4) I miss a short discussion on the data and methods used in the paper. For example, are there any limitations associated with the 1D mixed-layer model (see also comment to 1.158-159 below)? For the EGC, you mention that observations are biased toward summer (1.130). How is this for the northern AW branches? Any uncertainties with the atmospheric forcing (see comment below to 1.294)?

In the revised manuscript we have added a discussion on the limitations/uncertainties of the data and methods used (see details in the Method section). The seasonal data coverage biases are much smaller for the two NwAC branches.

Detailed comments:

1.30. I suggest you add a few more references here. Also, Chafik et al. (2021) is more of an opinion-paper published in EOS.

We have added Lozier et al. (2019), Keil et al. (2020) and Tsubouchi et al. (2021).

1.36-42. As mentioned, I think the authors should better establish the novelty of their analysis. Papers on the importance of lateral heat transfer from the NASC exist before 2021 (e.g., Isachsen et al. 2012). It is not clear to me why the authors choose to highlight an unpublished paper from 2022 (Renfrew et al; 1.38-42).

As we mentioned above, the manuscript has been revised to better establish the novelty of the study. In addition to citing other previous (published) studies, we felt it was worthwhile to reference the new paper by Renfrew et al. (soon to be accepted since the reviews were very favorable). By analyzing simultaneous observations of the atmosphere and ocean during a cold-air outbreak, Renfrew et al. highlighted the importance of lateral heat transfer on the transformation of AW. This is a particularly good example from an observational-based study, in addition to the modeling study of Isachsen et al. (2012).

1.48. The results from Smedsrud et al. (2022) show no significant changes in heat loss over the Nordic Seas. The decreased heat loss only takes place over the Greenland and Iceland Seas. Please comment.

For the centennial time scale (1900-2000), no significant change was found in heat loss over the Nordic Seas. However, for the time period of our study (1993-2018), there were significant changes.

1.49. “profound impacts” – please elaborate.

We have elaborated the impacts in the revised text.

1.54. “large-scale ramifications” – please elaborate.

We have elaborated on this in the revised text. For example, the re-organization of air-sea interaction due to the sea-ice retreat (Moore et al., 2022).

1.58. The Norwegian Atlantic Current is normally abbreviated NwAC to separate it from the North Atlantic Current (NAC).

The abbreviations have been updated.

1.96. It is not clear to me (from Methods) how the satellite observations and reanalysis product was combined (figure caption says “composite”).

The composite was done by averaging the GLORYS12 and satellite data on the same grid. This is now stated in the Methods section.

1.133-138. Why don’t you assess the roles of air-sea heat fluxes and lateral transport in the same way as for the other two branches? Basing this conclusion on measurements from one year is not convincing.

There is a thick and fresh Polar Surface Water layer at the surface of the EGC (upper 150m of water column, see gray shading in Figure 2c). The lateral displacement of this layer, which is highly variable and sensitive to the wind forcing, has an important impact on the transformation

of AW located below (see details in Huang et al., 2021). Since the PWP model does not consider such lateral processes, the quick formation of sea ice in model prevents the air-sea cooling of the AW layer. Therefore, it is not meaningful to use the same approach to assess the roles of air-sea heat fluxes on the transformation of AW along the EGC. We now state this limitation of the PWP model for the EGC in the Methods section.

l. 146-147: Please elaborate on these large differences and how your approach is different (and better, noting that you also make use of a model/reanalysis and a mixed-layer model).

As mentioned, the current 3D models (without data assimilation) do not accurately reproduce the NwAFC. In particular, the strength of NwAFC varies between models (e.g., Isachsen et al. 2012; Ypma et al., 2020; Spall et al., 2021; Treguier et al., 2021) and also between models and observations (see the observational results in Bosse et al., 2019). The GLORYS12 reanalysis assimilates several observational datasets and provides a relatively accurate estimate of the velocity of NwAFC. Therefore, we can combine this product with a 1D mixed-layer model to isolate the effect of air-sea cooling. We note that the heat budget is not balanced in GLORYS12, so that we are not able to completely quantify the roles of different lateral processes (e.g., lateral advection and diffusion) on the cooling of AW. We have elaborated on this in the revised manuscript.

l.158-159: “Differences between the simulated results and observations are assumed to be due to lateral processes.” How valid is this assumption? Could the differences be due to something else? Please elaborate.

The mixed layer depths simulated by the PWP model show good agreement with observations (the maximum mixed layer depth along the pathway is about 500 m). Since the lower boundary of the AW layer (550 m) is deeper than the maximum mixed layer depth, any vertical flux of heat at the

bottom of AW layer is negligible. Therefore, the differences between the PWP results and observations can be assumed to be due to lateral processes (e.g., lateral advection and diffusion). In the revised manuscript, we have explained this and have provided uncertainties in the model simulations and observations.

1.165. Figure 3b is presented before 3a. Reverse order?

The order has been revised.

1.191. Do conclusions depend on the chosen depth averaging? Would you expect results to change if you used a time-dependent mixed-layer depth? Also, earlier in the manuscript you used 550 m (Figure 2 and 3). Do these figures change for 0-200m?

Our conclusions do not depend on the chosen depth averaging (0-200 m). The mean mixed-layer depths along the two NwAC branches are shallower than 200 m. In addition, most of the temperature variability along the two NwAC branches is confined to the upper 200 m. Therefore, the 0-200m averaging is a good choice. The results are essentially the same if we used a time-dependent mixed-layer depth. The main findings in Figures 2 and 3 do not change for 0-200 m. This has been clarified in the revised manuscript.

1.194: “significant long-term reduction”. From Figure 4 it looks like there minimum AW cooling took place in the early 2000s after which it has increased. A similar development can be seen for temperature in the Norwegian Sea (Orvik 2022). I understand (and agree) that a linear trend is used in the analysis, but think the authors should make it clear in the text whether they interpret this as a long-term trend or (decadal) variability.

Thanks for the comment. Figure 4 also includes the signal of interannual variability, which was more pronounced before 2005. Future work is required to understand the mechanisms regulating this interannual variability. In this study, we interpret the change from 1993 to 2018 as a long-term trend. This has been clarified in the revised manuscript.

1.233-236. Wouldn't lateral heat transfer depend somewhat on the alongstream velocity?

This might be the case. For instance, it could affect the ability of the flow to be barotropically unstable. This would be interesting to pursue, but is beyond the scope of our study.

1.236-238/1.258. Could the authors speculate/discuss how the identified difference in water mass transformation along the different AW pathways could impact the response of the high-latitude AMOC to a changing climate? Would this make the AMOC more or less sensitive to change? And would this imply that coarse resolution climate models that don't resolve the NAFC are more/less sensitive to climate change?

This is now addressed in the last paragraph of the revised manuscript, but we do not come to any clear conclusions. We note that the fate of the AW in the two NwAC branches is very different: while much of the transformed AW transported by the NwAFC returns southward via the EGC as part of the AMOC, a significant portion of the AW in the NwASC flows northward into the Arctic Ocean via the West Spitzbergen Current and to the Barents Sea via the North Cape Current. These latter two currents contribute to the overturning in the high Arctic. Recent studies have demonstrated that the AW in the EGC, as well as that in the boundary current north of Svalbard in the Amundsen Basin, can be further transformed as a result of reduced ice cover in these regions (e.g., Våge et al, 2018; Huang et al., 2021; Perez-Hernandez et al, 2019). Hence, it remains unclear how much the reduction in air-sea cooling along the NwAFC might be compensated for by the re-ventilation along the EGC. This makes it challenging to predict how the AMOC and Arctic

overturning will respond to the differing effects of air-sea heat exchange on the two NwAC branches. We conclude the last paragraph by saying that future observational and modeling studies are required to address this.

1.243. “The densification of AW was found to occur...” This phrasing makes it sound like this is a new finding of this paper. Suggest rephrasing.

It has been rephrased.

1.260-265. I suggest that the authors rather highlight the new findings (novelty) of this study rather than ending with all that remains to be understood.

Thanks for the suggestion. We have highlighted the new findings (novelty) in the “Conclusion and Discussion” section.

1.294. Are your results sensitive to the chosen reanalysis product? Temporal variability can be quite different for different products (e.g., Carton et al. 2011).

In addition to the GLORYS12, we also looked at the ASTE reanalysis product, which has high resolution and covers the Nordic Seas. However, we found that the NwAFC is not well represented in the ASTE. This problem also exists in two other available reanalysis products (ARMOR3D and GREP), whose spatial resolutions are more coarse ($1/4^\circ$ and 1° , respectively). Furthermore, studies (e.g., Lellouche et al. 2021) have documented that the GLORYS12 nicely captures the low frequency variability (interannual and decadal). This has now been clarified in the Methods section.

Reviewer #3 (Remarks to the Author):

The paper is clearly written, logically presented, and have important results. It suggests that the cooling of the NAFC is reduced over the period 1993-2018. In contrast, changes in cooling are less clear in the NASC where advection play a bigger role. Before possible publication there are some major points that the authors should address. This is especially related to the validity of using a time-constant pathways that for the NAFC branch that may lead to methodological problems. If this can be sorted out convincingly this paper would be a valuable contribution with both regional and larger scale interest.

Major points

1. Fig1b now basically show the extent of the Atlantic layer but avoiding the Lofoten Basin since the lower interface of the Atlantic Layer here is deeper, typically at 800m. Using the 550-600 m layer for defining the pathways, minimum in density-spiciness space, is not justified. Please explain.

The pathways of AW in this study were determined from surface velocity field. The 550-600 m is the only depth range where the distribution of minimum sigma-pi distances shows good agreement with the determined pathways. At the depth below 600 m, the minimum distances are isolated inside the Lofoten Basin, with no clear connection to the two NwAC branches. Since our focus in this study is on the transformation of AW along the boundary currents, using 550-600m as the lower interface of AW layer is justified. We have clarified this in the revised manuscript.

2. The hydrographic database used here is from 1993-2018. Why do limit to the period 2005-2018 when defining the pathways, sigma-spiciness distance. Would they look different during the early period? The reason for bringing this up is that while the slope current is basically trapped by the

shelf break, the NAFC may have larger potential for zonal shift (see e.g. Blindheim et al., 2000), that also may vary meridionally. You assume a fixed pathway in this study. However, can you rule out that your reduced cooling in the NAFC is not a result of varying zonal extent of the Atlantic Water? Figure 4 suggest that the reason for the different cooling trends in the Front and Slope currents are basically explained by the “cold” water in the northern part of the NAFC in the early part (before 2004). The authors need to provide convincing arguments that the conclusion is not a result of an unrealistic fixed pathway for the NAFC, since it is in the core of the main result and conclusion of this paper.

The reason we consider the period 2005-2018 to investigate the overall AW transformation in the two currents is to avoid the impact of long-term variability on the water mass transformation, which can impact the results of the sigma-pi distance. With regard to the temporal variability of the pathways, the new Supplementary Figure 5a shows the yearly pathways of the two NwAC branches defined by the maxima in the surface velocity field in each year. While the pathways of NwAFC are more variable than that of NwASC, there is no clear long-term change in the pathway of either current. Since the width of each current is defined to be a 50 km swath, our results concerning the long-term changes of net cooling of AW, air-sea heat flux, and along-stream velocity are not sensitive to the yearly variation of the pathways (see Supplementary Figure 5b-g). This is explained in the revision.

Minor points:

1. References: Roshin 2018,2019 should be Raj;

Thanks for catching this. The references have been corrected.

2. Fig 1c, difficult to see if there is a maximum in velocity (speed), especially for the NAFC but also partially for the NASC.

In revised Figure 1 we only show the velocities larger than 0.04 m/s. The pathways and west-east boundaries of the pathways (25 km on either side of the velocity maxima) are presented more clearly now.

3. Please provide the argument why reduced cyclonic wind forcing should increase the NAFC.

In the revised manuscript, we have provided more explanation about the long-term change in the velocity of the two NwAC branches. Raj et al. (2018, 2019) suggested that the strength of the NwAFC (NwASC) increases (decreases) during negative NAO periods, when the cyclonic wind forcing over the Nordic Seas weakens. This is the reason we felt that the long-term changes for the alongstream velocity of two NAC branches were likely associated with the changes of wind forcing. However, we have now carefully analyzed the ERA5 reanalysis data which revealed only a slight long-term weakening of cyclonic wind forcing in the Greenland Sea, and no such trend in the eastern Nordic Seas. This implies that wind forcing is not driving the long-term change of circulation for the two NwAC branches.

To better understand this, we compared the changes of sea surface height (SSH) between two time periods: 2000-2002 and 2014-2016. We note that both time periods are not associated with extreme NAO events, but there are significant changes with respect to net cooling of AW, air-sea heat flux, and surface velocity along NwAFC (see revised Figure 4 and the new Supplementary Figure 6). The SSH change map provides an explanation for the long-term change in the circulation along NwAFC. In particular, the increase of SSH is stronger in the eastern Nordic Seas, especially in the Lofoten Basin. This change leads to an SSH gradient across the NwAFC which intensifies the strength of current. Such a long-term change in the SSH field has also been found in Broomé et al.

(2020), who suggested that this change was dominated by the long-term heat content change in the Nordic Seas due to warming of the inflowing AW water. In summary, while the wind forcing can play a role in the circulation change during the extreme NAO years (i.e., interannual variability), the long-term change in the velocity field seems to be associated with the asymmetric heat content change in the Nordic Seas. We have explained all of this in the revision.

REVIEWER COMMENTS

Reviewer #1 (Remarks to the Author):

The authors have addressed to full satisfaction my comments, and those of the other two reviewers. On reading the revised manuscript I do however have two further comments:

(1) On p.10, lines 203 and 207 (also in the captions of Figs. 4 and 5, and elsewhere), the authors explicitly refer to (or imply) the use of mean air-sea turbulent heat flux. Strictly speaking, the turbulent heat flux comprises the sensible and latent components, but not the net longwave and shortwave components. There is no reason why reduced net longwave or increased shortwave radiation (through e.g., changing cloud cover) could explain some of the AW cooling. Can the authors clarify what part of the net heat flux they are using, and if only the actual turbulent components, please justify the neglect of radiative components.

(2) In Methods, 'One-dimensional mixing model following the current', are surface freshwater fluxes (E-P) excluded? If so, please justify this exclusion.

Reviewer: Professor Robert Marsh

Reviewer #2 (Remarks to the Author):

I thank the authors for their detailed response to my questions and comments. My concerns have been satisfactorily addressed and I have only a few additional minor comments.

I.36-41: I don't think it's correct, as a general statement, to say that the importance of lateral mixing has only been demonstrated "recently" and then to cite two papers from their own group. Although previous papers have had a more regional focus (e.g., on the Lofoten Basin), they are still (in part) about lateral mixing of the NwAC.

Some examples (full references below):

Richards and Straneo (2015)

Bosse et al. (2018)

Nilsen et al. (2006), and references therein, for the West Spitsbergen Current.

Isachsen et al (2012) also includes observations (title of the paper: "Observed and modeled surface eddy heat fluxes in the eastern Nordic Seas").

I.58-60: As you focus mainly on the two NwAC branches, could you elaborate on a "large-scale ramification" that involves these branches and not the EGC (focus of Moore et al. 2022)?

I.115: typo – Mauritzen

I.143-144: Could you specify what "large differences" mean and whether they were identified by you? This also related to I.337-338 – what other reanalysis did you consider and is there a figure or reference showing this comparison? And is there a reason for why GLORYS12 doesn't have the same problem of accurately representing the NwAC even if the horizontal resolution is the same as e.g., Treguier et al.?

I.179: typo – Isachsen.

I.229-230: Could you comment on how your inferred weakening (strengthening) of the NwASC (NwAFC) relates to observed changes in AW inflow (Tsubouchi et al. 2021) and from the NwASC (Orvik 2022), the latter reporting no long-term trend between 1995-2020.

I.304: "as part of the AMOC". Isn't AW transformation in the Arctic also part of the AMOC? Suggest to rephrase.

I.305: Spitsbergen

I.307: Atlantification of the Barents Sea was first introduced by Årthun et al. (2012). Lind et al mainly reports weakened stratification in the northwestern Barents Sea as a result of reduced ice import.

References:

Årthun et al. (2012). <https://doi.org/10.1175/JCLI-D-11-00466.1>

Nilsen et al. (2006). <https://doi.org/10.1029/2005JC002991>

Orvik (2022). <https://doi.org/10.1029/2021GL096427>

Richards and Straneo. <https://doi.org/10.1175/JPO-D-14-0238.1>

We appreciate the reviewers' comments and suggestions on the manuscript. Our responses below are in blue font.

Reviewer #1 (Remarks to the Author):

The authors have addressed to full satisfaction my comments, and those of the other two reviewers. On reading the revised manuscript I do however have two further comments:

(1) On p.10, lines 203 and 207 (also in the captions of Figs. 4 and 5, and elsewhere), the authors explicitly refer to (or imply) the use of mean air-sea turbulent heat flux. Strictly speaking, the turbulent heat flux comprises the sensible and latent components, but not the net longwave and shortwave components. There is no reason why reduced net longwave or increased shortwave radiation (through e.g., changing cloud cover) could explain some of the AW cooling. Can the authors clarify what part of the net heat flux they are using, and if only the actual turbulent components, please justify the neglect of radiative components.

Thanks for the comment. For the long-term mean PWP simulation (results section 3), we used the total air-sea heat flux, including both turbulent and radiative components. Regarding the results associated with the long-term change of air-sea heat flux (results section 4), we only used the turbulent components, whose change dominates the change of total heat flux. The long-term change of radiative components over the two NwAC branches is negligible (integrated change from 1993 to 2018 is smaller than 2 W/m^2). This has been clarified in the revised manuscript.

(2) In Methods, 'One-dimensional mixing model following the current', are surface freshwater fluxes (E-P) excluded? If so, please justify this exclusion.

The surface freshwater fluxes (E-P) are excluded. The air-sea heat fluxes provide the dominant contribution to mixed-layer deepening in the PWP simulations of the Nordic Seas (e.g., Våge et al. 2018). The impact of freshwater fluxes on the temperature change of AW are negligible. This has been clarified in the revised manuscript.

Reviewer #2 (Remarks to the Author):

I thank the authors for their detailed response to my questions and comments. My concerns have been satisfactorily addressed and I have only a few additional minor comments.

l.36-41: I don't think it's correct, as a general statement, to say that the importance of lateral mixing has only been demonstrated "recently" and then to cite two papers from their own group. Although previous papers have had a more regional focus (e.g., on the Lofoten Basin), they are still (in part) about lateral mixing of the NwAC.

Some examples (full references below): Richards and Straneo (2015); Bosse et al. (2018); Nilsen et al. (2006), and references therein, for the West Spitsbergen Current; Isachsen et al (2012) also includes observations (title of the paper: "Observed and modeled surface eddy heat fluxes in the eastern Nordic Seas").

Thanks for the comment. We have rephrased the statement and cited the suggested references.

l.58-60: As you focus mainly on the two NwAC branches, could you elaborate on a "large-scale ramification" that involves these branches and not the EGC (focus of Moore et al. 2022)?

We have elaborated on this in the revised text. Kiel et al. (2020) provides a large-scale ramification that involves the two NwAC in the Nordic Seas. They demonstrated that the changes of the high-latitude heat transport by the boundary currents play an important role in the cooling of North Atlantic warming hole.

l.143-144: Could you specify what "large differences" mean and whether they were identified by you? This also related to l.337-338 – what other reanalysis did you consider and is there a figure or reference showing this comparison? And is there a reason for why GLORYS12 doesn't have the same problem of accurately representing the NwAC even if the horizontal resolution is the same as e.g., Treguier et al.?

Thanks for the comment. In the revised manuscript, we have now specified the large differences and pointed out where they were identified (e.g., by comparing Figure 2c in Isachsen et al. 2012, Figure 5 in Treguier et al. 2021, and Figure 2 in Bosse and Fer 2018).

We also compared the long-term mean (2005-2018) sea surface velocity along the NwAFC from GLORYS12 (1/12°) with three other reanalysis products: ARMOR3D (1/4°), ASTE (15-20 km), and GREP (1°) (see details of each product in supplementary Table 1). This revealed that ARMOR3D seems to do a comparable job to GLORYS12 in capturing the current, but the other two products are deficient in this regard. We now mention this in the revised text, but we don't think it warrants a figure.

It seems likely that in high-resolution prognostic models with limited assimilation of observational datasets (e.g., Treguier et al. 2021), the NwAFC is not well reproduced. This problem also exists in the ASTE reanalysis product where a reversal of NwAFC was found along the Mohn Ridge. This has been clarified in the revised manuscript.

1.229-230: Could you comment on how your inferred weakening (strengthening) of the NwASC (NwAFC) relates to observed changes in AW inflow (Tsubouchi et al. 2021) and from the NwASC (Orvik 2022), the latter reporting no long-term trend between 1995-2020.

We did discuss that the strengthening of NwAFC can be associated with the asymmetric steric change in the SSH of the Nordic Seas, which is relative to the long-term heat content change from the AW inflow (see more details in the third paragraph of results section 4). Regarding the long-term observation of NwASC at the Svinøy line (Orvik 2022), we do not come to any clear answer of how it relates to our results. As we seen from Figure 5e, the sign of long-term trend in the strength of NwASC can change from negative to positive at different latitudes.

1.304: “as part of the AMOC”. Isn't AW transformation in the Arctic also part of the AMOC? Suggest to rephrase.

It has been rephrased to “as part of the subpolar AMOC”.

1.307: Atlantification of the Barents Sea was first introduced by Årthun et al. (2012). Lind et al mainly reports weakened stratification in the northwestern Barents Sea as a result of reduced ice import.

Thanks for the suggestion. We have now cited Årthun et al. (2012).

1.115: typo – Mauritzen

1.179: typo – Isachsen.

1.305: Spitsbergen

Done.